# SEHG: Bridging Interpretability and Prediction in Self-Explainable Heterogeneous Graph Neural Networks

## Abstract

Heterogeneous Graph Neural Networks (HGNNs) are extensively applied in modeling web-based applications that involve heterogeneous graph structures. Explanation models for HGNNs aim to address their "black box" nature. Enhancing the interpretability of HGNNs leads to a better understanding and can potentially improve predictive performance. However, existing post-hoc HGNN explanation methods cannot impact the HGNN's predictions. Self-explainable homogeneous models also perform poorly on heterogeneous graphs. To address these challenges, we present a Self-Explainable Heterogeneous Graph Neural Network (SEHG), a novel architecture that integrates explanation generation into the learning process of HGNN through two alternative stages. The first stage focuses on producing high-quality explanations while providing predictions alongside. The second stage enhances prediction accuracy by a contrastive learning strategy. Unlike the current methods that rely on manually defined metapaths for structural explanations, SEHG generates important structure and feature explanations by learnable heterogeneous masks. To ensure high-quality and sparsity explanation, these masks are regulated by a uniquely designed range-based penalty during training. Moreover, we introduce HetBA, a collection of synthetic heterogeneous datasets designed to quantify and visualize explanations or heterogeneous graphs. Extensive experiments demonstrate the effectiveness of SEHG, which surpasses strong baselines in real-world node classification tasks by notable margins of up to 3.91%. SEHG also achieves state-of-the-art performance on synthetic datasets with improvement of up to 9.44%, and records the highest fidelity scores in explanation tasks, improving by up to 46.57%. To our knowledge, SEHG is a pioneering self-explainable HGNN framework that achieves state-of-the-art performance on both heterogeneous graph explanation and prediction tasks.

## CCS Concepts

• **Computing methodologies → Machine learning**.

## Keywords

Heterogeneous Graph Neural Network, Graph Explanation, Self-Explainable, Graph Self-Supervised Learning

**ACM Reference Format:**
Anonymous Author(s). 2025. SEHG: Bridging Interpretability and Prediction in Self-Explainable Heterogeneous Graph Neural Networks. In *Proceedings of the ACM Web Conference 2025 (WWW '25)*. ACM, New York, NY, USA, 13 pages. https://doi.org/XXXXXXX.XXXXXXX

## 1 INTRODUCTION

Heterogeneous relationships, characterized by diverse node and edge types, are inherent in many real-world systems and web applications and can be presented as heterogeneous graphs. Heterogeneous Graph Neural Networks (HGNNs) [23] are effective models to process heterogeneous graphs and are applied in various contexts [4], including social networks [4, 31], and anomaly detection [11, 44]. While numerous advanced HGNNs have been proposed [10, 24, 34, 36, 42], these methods primarily focus on enhancing model performance by specialized architectures and exploitation of heterogeneous data characteristics. However, traditional HGNNs do not actively elucidate the model's fundamental understanding of the data, instead, they function as black boxes, similar to many deep learning methods [18], obscuring the mechanisms underlying the learned representations in HGNNs.

Approaches such as HGExplainer [27] and HTGExplainer [19], xPath [20], inspired by explainability methods in homogeneous graphs [12, 30, 38, 41], have been developed to explain predictions in heterogeneous graphs in a post-hoc manner after the HGNN is well trained and remains unchanged. While these post-hoc methods provide valuable insights into the HGNN's decision-making, the HGNN's predictions remain unchanged from the explanations. HGNNs might still learn irrelevant information that is not reasonable from explanations from original data even after the training is finished, leading to sub-optimal predictions. Based on the sub-optimal predictions, the explainers can also produce unsatisfactory explanations. Even though self-explainable homogeneous graph neural networks like SEGNN [6], GSAT [26], ProtGNN [43], SES [13], and ExpFiGCN [35] exist, they fail to produce satisfactory explanations and predictions due to the huge gap between the characteristics of heterogeneous and homogeneous graphs.

To address the challenges in explaining HGNNs, we present a **S**elf-**E**xplainable **H**eterogeneous **G**raph Neural Network (SEHG), which integrates heterogeneous explainability into the training process of HGNNs. SEHG consists of two primary phases: heterogeneous self-explainable training and enhanced contrastive learning. In the heterogeneous self-explainable training phase, we proposed a novel heterogeneous graph explanation generator that produces node feature and edge structure masks. According to the previous study [40], high-quality explanations should be sparse, capturing the most important input features, while disregarding the irrelevant ones. To ensure this sparsity, we incorporate specially designed sparsity-enhancing penalty terms. In the enhanced contrastive learning phase, SEHG extracts appropriate sample pairs from the explanation graph for each predicted node to facilitate contrastive

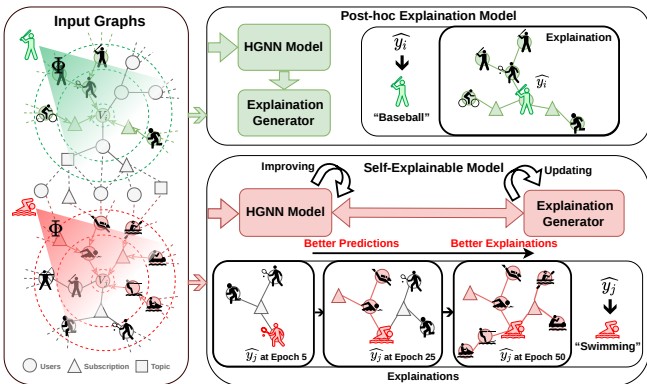

**Figure 1: An illustration of heterogeneous post-hoc explanation models and the SEHG mechanism, demonstrated in a hypothetical node classification task. A GNN $\Phi$ is trained on heterogeneous social networks to predict the future sports activities of the center nodes. Post-hoc explanation models provide explanations after the HGNN is trained, whereas SEHG enhances HGNN predictions through integrated explanations, with improved predictions further refining the quality of explanations throughout the training process.**

learning. Inspired by SimCLR [5], a learnable nonlinear transformation is introduced between representation and contrastive losses, significantly enhancing the quality of the learned representations. The difference between SEHG and post-hoc explanation methods such as xPath [20] and HGExplainer [27] is illustrated in Fig. 1. Post-hoc explanation models explain a trained HGNN without impacting its performance, while SEHG simultaneously enhances predictions and explanation quality during training.

Existing metrics for quantifying the heterogeneous explanation models primarily rely on fidelity assessments using available datasets [40] lack intuition visualization of explanation quality and reliable ground truth for accurately and fairly evaluating explanation accuracy. Inspired by GNNExplainer [38], we introduce a group of synthetic datasets **HetBA** to evaluate and visualize the explainability performance of heterogeneous models. HetBA consists of four subsets—SingleShapes, DoubleShapes, TripleShapes, and TripleCommunities—each featuring planted heterogeneous network motifs as indicators of ground truth. This design enables a comprehensive and intuitive evaluation and visualization of various explanation methods.

We evaluated the predictive performance and explainability of SEHG on four real-world heterogeneous datasets and four synthetic datasets. SEHG consistently achieves the highest scores across all datasets. Notably, SEHG demonstrated up to a 3.91% improvement in prediction performance measured by Micro-F1 and a significant 46.57% enhancement in explanation quality assessed by fidelity scores. SEHG also achieves state-of-the-art performance on the HetBA dataset with an improvement of up to 9.44%. Visualization of the explanation results on HetBA also confirms that SEHG accurately identifies target structures, demonstrating its superior ability to pinpoint and leverage essential features. Moreover, SEHG is compatible with various backbone HGNNs, illustrating that the prediction performance and explanation quality can be optimized within a single framework.

Our contributions are as follows:

- We present SEHG, a robust self-explanation HGNN framework that integrates a unique explanation generator into the HGNN training pipeline and effectively leverages explanations during enhanced contrastive learning, improving performance in both explanation and prediction.
- We proposed a range-based penalty method to ensure the sparsity of explanations, preventing ambiguity and ensuring high-quality explanations.
- We introduce HetBA, a group of datasets specifically designed for heterogeneous graph explanation tasks that allow comprehensive evaluation and benchmarking of explanation methods across various scenarios.
- Extensive experiments demonstrate that SEHG achieves state-of-the-art performance in explanation and prediction tasks, highlighting the superiority of its self-explanation mechanism and contrastive learning framework in heterogeneous graphs.

## 2 RELATED WORKS

### 2.1 Heterogeneous Graph Neural Networks

HGNNs have exhibited robust representation capabilities for managing heterogeneous graph data [2, 33]. These models extend traditional graph neural network (GNN) frameworks [9, 14, 15] to handle complex, multi-relational data by leveraging diverse types of nodes and edges. For instance, HAN [34] preserves graph heterogeneity by constructing neighbors through meta-paths, and predefined sequences of node and edge types, enabling it to capture diverse relational information. In contrast, GTN [28] learns node representations by aggregating features from neighbors without predefined meta-paths, offering greater flexibility. SHGAT [24] is a simplified HGNN variant based on GAT, fine-tuned for robust performance. SeHGNN [36] uses a lightweight mean aggregator for pre-computation, integrating meta-path features with a single-layer transformer, enhancing receptive fields and feature integration. HetGNN [42] samples correlated heterogeneous neighbors through a random walk and aggregates features using a two-module neural network. Finally, HGT [10], using a transformer-based architecture and subgraph sampling, boosts scalability for large web graphs.

However, the "black box" nature of HGNNs poses challenges, particularly in areas where transparency is key for fairness, trust, and accountability [18]. This lack of interpretability complicates the understanding of model predictions, raising concerns about bias and reliability. In sensitive fields like recommendation systems and user behavior analysis, this opacity undermines trust, hinders discrimination detection, and increases ethical risks [8, 32].

### 2.2 Explanation of Graph Neural Networks

To provide explanations for GNNs, several methods and explainers have been proposed [40]. These explainers are primarily divided into post-hoc and self-explainable models. Among post-hoc explainers, GNNExplainer [38] is a pioneering instance-level model that elucidates edge and feature importance by maximizing mutual information between GNN predictions and subgraph structures. PGExplainer [22] uses a deep neural network to generate multi-instance explanations, while GraphLIME [12] offers local interpretability through HSIC Lasso, focusing on feature explanations.

XGNN [39], a model-level approach, employs a graph generator to optimize specific predictions but may lead to discrepancies between the explainable model and the GNN. In contrast, self-explainable GNNs provide explanations during training. SEGNN [6] uses an interpretable similarity module to identify K-nearest labeled nodes for explainable node classification. ProtGNN [43] integrates pro-totype learning for case-based explanations. GSAT [26] applies stochasticity to attention weights to filter irrelevant graph components and highlight task-relevant subgraphs. ExpFiGCN [35] selects key nodes while denoising, enhancing node representation in GCN, and addressing over-smoothing, while SES [13] extends this self-explanation approach to any GNN and various datasets.

Research on explainable models for heterogeneous graphs is still limited. HGExplainer [27] accounts for temporal dependencies while preserving heterogeneity in subgraph explanations. HTGEx-plainer [19] improves heterogeneity representation by maximizing joint mutual information and using meta-path-based sampling for richer insights. xPath [20] efficiently identifies influential node pairs through graph perturbation, using a greedy search algorithm to pinpoint the most impactful fine-grained explanations. However, these approaches operate independently from the core learning process, leading to unsatisfactory predictions, and their explanations do not contribute feedback to the training of HGNNs.

## 3 PRELIMINARIES

### 3.1 Heterogeneous Graph Neural Networks

**Heterogeneous Graph Definition:** A heterogeneous graph is defined as $G = (V, H, E, \mathcal{T}_v, \mathcal{T}_e)$, where $V = \{v_1, v_2, \ldots, v_n\}$ represents the set of nodes, containing different node types. $H = \{h_1^{(0)}, h_2^{(0)}, \ldots, h_n^{(0)}\}$ represents the set of initial features, where $h_i^{(0)}$ is the feature vector for node $v_i$. $E = \{e_1, e_2, \ldots, e_m\}$ denotes the set of edges, containing different edge types. $\mathcal{T}_v$ is a node-type mapping function that assigns each node to a specific type. $\mathcal{T}_e$ is an edge-type mapping function that assigns each edge to a specific type.

**Message Passing Mechanism:** At layer $l$, the representation of node $v_i$ is updated by aggregating messages from its neighboring nodes $\mathcal{N}(v_i)$ as follows:

$$\mathbf{h}_i^{(l)} = \sigma \left( \sum_{r \in \mathcal{T}_e(E)} \sum_{v_j \in \mathcal{N}_r(v_i)} \frac{1}{c_{ij}} \mathbf{W}_r^{(l)} \mathbf{h}_j^{(l-1)} + \mathbf{W}_0^{(l)} \mathbf{h}_i^{(l-1)} \right), \quad (1)$$

where $\mathcal{N}_r(v_i)$ denotes the set of neighboring nodes connected to $v_i$ by edges of type $r$, $\mathbf{W}_r^{(l)}$ and $\mathbf{W}_0^{(l)}$ are the weight matrices for type $r$ edges and self-loops, respectively, $c_{ij}$ is a normalization factor, and $\sigma(\cdot)$ is an activation function, such as ReLU.

**Final Representation and Task:** After $L$ layers of message passing, the final representation of each node $h^{(L)i}$ is obtained. This representation can be used for various tasks, such as node classification, link prediction, or graph classification.

### 3.2 Heterogeneous Graph Explanation

Given a heterogeneous graph $G$ and a trained HGNN model $f : G \rightarrow \mathbb{R}^C$, where $C$ is the number of classes or prediction outputs, the goal of an explanation is to identify the most influential components of $G$, represented by $G_e = (V_e, H_e, E_e) \subseteq G$, contribute to the model's prediction $\hat{y}_i = f(G, v_i)$ for a given node $v_i \in V$.

## 4 SELF-EXPLAINABLE HETEROGENEOUS GRAPH NEURAL NETWORK

In this section, we provide a detailed description of SEHG, including heterogeneous self-explainable training and enhanced contrastive learning, as illustrated in Fig. 2. The normalized heterogeneous graph input generates node feature and edge structure masks, forming an explanation graph. This graph is processed through an HGNN encoder, where feature loss is constrained by range penalty loss alongside prediction losses. During the enhanced contrastive learning phase, sample nodes are processed through heterogeneous graph convolutional and linear layers to produce positive and negative representations for contrastive learning. This approach optimizes the final explanation and enhances prediction accuracy.

### 4.1 Heterogeneous Self-Explainable Training

As the first component of SEHG, Heterogeneous Self-Explainable Training normalizes the input graph features and employs a heterogeneous graph explanation generator to produce node feature masks and edge structure masks, which are optimized alongside the HGNN during the training process.

**Heterogeneous Representation Normalization.** Given a node $v_i$ with an initial feature vector $h_i^{(0)} \in \mathbb{R}^{d_r}$, where $d_r$ is the dimension of $h_i^{(0)}$, depending on its type $r \leftarrow \mathcal{T}_v(v_i)$, the transformation through two linear layers to output a final feature vector $h_i \in \mathbb{R}^{d_{\text{out}}}$ is expressed as:

$$h_i = \sigma \left( \mathbf{W}_2 \cdot \sigma(\mathbf{W}_1^r \cdot h_i^{(0)} + \mathbf{b}_1) + \mathbf{b}_2 \right), \quad (2)$$

where $\mathbf{W}_1^r \in \mathbb{R}^{d_{\text{hid}} \times d_r}$ is the weight matrix of the first linear layer specific for nodes belong to type $r$, $d_{\text{hid}}$ is the dimension of the hidden layer, $\mathbf{b}_1 \in \mathbb{R}^{d_{\text{hid}}}$ is the bias vector of the first layer, $\mathbf{W}_2 \in \mathbb{R}^{d_{\text{norm}} \times d_{\text{hid}}}$ is the weight matrix of the second linear layer, $d_{\text{norm}}$ is the dimension of normalized node feature, $\mathbf{b}_2 \in \mathbb{R}^{d_{\text{norm}}}$ is the bias vector of the second layer, and $\sigma(\cdot)$ is an activation function applied after each linear transformation, $h_i \in \mathbb{R}^{d_{\text{norm}}}$ is the final output feature vector. This transformation normalizes the heterogeneous features into a common representation space to obtain $H_{\text{norm}} \in \mathbb{R}^{N \times d_{\text{norm}}}$, where $N$ is the total nodes number. This normalization facilitates subsequent graph processing tasks.

**Heterogeneous Graph Encoder.** The heterogeneous graph encoder forms the basis for both explanation and prediction, and it can be substituted by any HGNN. To improve performance, we incorporate edge aggregation into the backbone model. For illustrative clarity, we provide specific details using SHGAT [24] as the backbone example.

The following formula aggregates the edge attribute:

$$h_{e_{ij}} = \mathbf{W}_e \cdot [h_i \parallel h_j \parallel \mathcal{T}_e(e_{ij})], \quad (3)$$

$\mathbf{W}_e \in \mathbb{R}^{(2 \times d_{\text{norm}} + 1) \times d_{\text{hid}}}$ is the learnable weight matrix that transforms the concatenated features into the edge feature $h_{e_{ij}}$, and $\parallel$ denotes the concatenation operation along one dimension. The

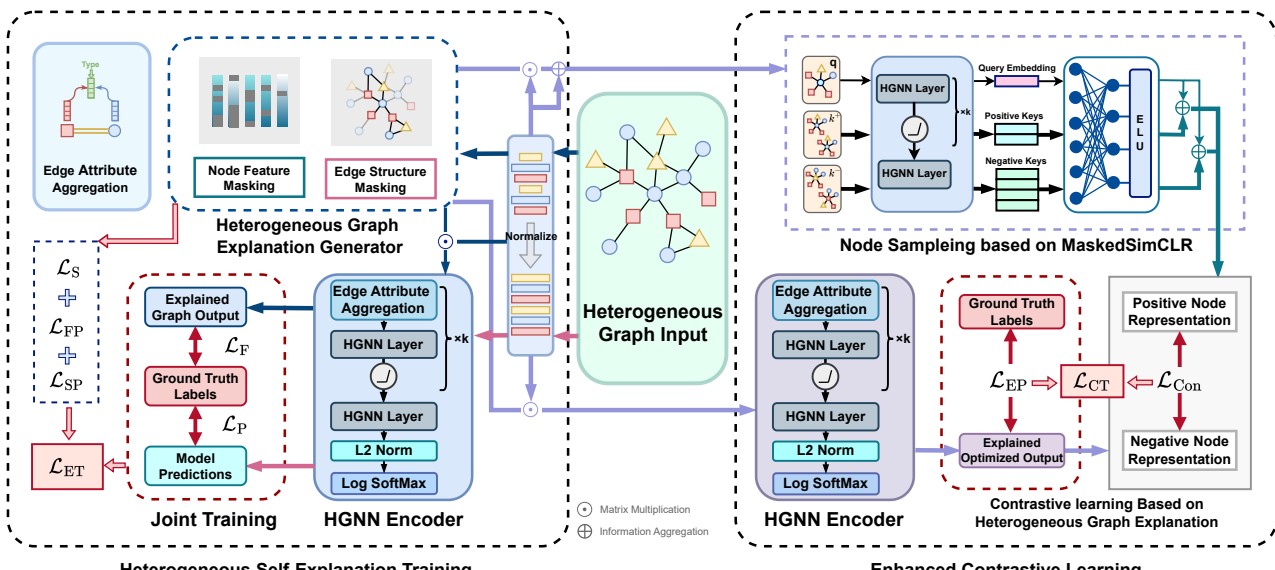

**Figure 2: The framework of SEHG, includes two main components: heterogeneous self-explainable training and enhanced contrastive learning. Different colored lines are used to represent distinct data processing workflows.**

weight $\alpha_{ij}$ for each edge $e_{ij}$ is from an attention mechanism as:

$$\alpha_{ij} = \frac{\exp\left(\text{LeakyReLU}(a^T[\mathbf{W}h_i\|\mathbf{W}h_j\|h_{e_{ij}}])\right)}{\sum_{k\in\mathcal{N}(v_i)}\exp\left(\text{LeakyReLU}(a^T[\mathbf{W}h_i\|\mathbf{W}h_k\|h_{e_{ij}}])\right)}, \quad (4)$$

where $a$ and $\mathbf{W}$ are learnable weights and $\mathcal{N}(v_i)$ represents the neighbors of node $i$. The $l$-th HGNN layer can be expressed as:

$$h_i^{(l)} = \sigma\left(\sum_{v_j\in\mathcal{N}(v_i)}\alpha_{ij}^{(l)}\mathbf{W}^{(l)}h_j^{(l-1)} + \mathbf{b}^{(l-1)}\right), \quad (5)$$

where $\alpha_{ij}^{(l)}$ is the attention weight about edge $e_{ij}$ and $\mathbf{W}^{(l)}$ is the weight of layer $l$.

**Heterogeneous Graph Explanation Generator.** The heterogeneous graph explanation generator is employed to generate learnable node feature masks and structural masks.

After passing through $k$ layers of the HGNN Layer, followed by a linear layer and an ELU activation, we obtain the node feature mask $M_f \in \mathbb{R}^{N\times d_{\text{norm}}}$. The mask is multiplied with the normalized representation $H_{\text{norm}}$ to generate the node feature explanation, which is then used to construct the edge structure explanation. For each node in the graph, we perform edge aggregation (Equation 3) by computing the representations of edges directly connected to it (within its one-hop neighborhood) and the edge representations of nodes within its two-hop neighborhood that are not directly connected. These real and virtual edge representations enhance the discriminative power of the linear layer. Specifically, for a given node $v_i$, we calculate the edge strengths within its two-hop neighborhood:

$$m_{e_{ij}} = \mathbf{W}_m\cdot\left(\mathbf{W}_e\cdot[h_i\|h_j\|\mathcal{T}_e(e_{ij})]\right)+\mathbf{b}_m \quad (v_j\in\mathcal{N}_2(v_i)), \quad (6)$$

where $\mathbf{W}_m\in\mathbb{R}^{d_{\text{hid}}\times1}$ is used to compute the edge strength, and $b_m\in\mathbb{R}^1$ is the bias term. $\mathcal{N}_2(v_i)$ is the 2-hop neighbor nodes of $v_i$ and $\mathcal{T}_e(e_{ij})$ is set to $-1$ if $e_{ij}$ does not exists. All edge strengths $m_{e_{ij}}$ are combined to form the structural mask $M_s\in\mathbb{R}^{N\times N}$, where $i$ and $j$ denote the row and column indexes in $M_s$.

For the structure mask $M_s$, we introduce a threshold-based penalization loss $\mathcal{L}_{\text{SP}}$ constraint. Specifically, the elements of $M_s$ are categorized into three segments (greater: $V_{s>}$, middle: $V_{s\approx}$, less: $V_{s<}$, each subjected to distinct penalties:

$$\begin{aligned} V_{s>} &= M_s\cdot\mathbb{I}(M_s > 2\delta_{\text{avg}}), \\ V_{s\approx} &= M_s\cdot\mathbb{I}(\delta_{\text{avg}} < M_s \le 2\delta_{\text{avg}}), \\ V_{s<} &= M_s\cdot\mathbb{I}(M_s \le \delta_{\text{avg}}), \end{aligned} \quad (7)$$

where $\delta_{\text{avg}}$ is the threshold used to segment the regions, with its value typically set between 0 and 0.5. These three segments are used to calculate structure explanation penalization loss $\mathcal{L}_{\text{SP}}$:

$$\mathcal{L}_{\text{SP}} = \lambda_{\text{L1}}\cdot\sum_{m\in M_s}|m| + \left|(\delta_{\text{avg}}^2\cdot d_{\text{norm}}) - \sum_{i=1}^{N}\left(V_{s>}^i + V_{s<}^i - V_{s\approx}^i\right)\right|, \quad (8)$$

where $\lambda_{\text{L1}} = \frac{0.01}{\delta_{\text{avg}}^2\cdot N_p\cdot d_{\text{norm}}}$. This feature explanation loss incorporates an L1 regularization term to ensure sparsity and a range-based penalty to enhance the discriminability of $M_s$. The penalty term $\left|(\delta_{\text{avg}}^2\cdot d_{\text{norm}}) - \sum_{i=1}^{N}(V_{s>}^i + V_{s<}^i - V_{s\approx}^i)\right|$ is designed to achieve two main objectives: 1) Encouraging Extremes: By maximizing the sum of $V_{s>}$ and $V_{s<}$ while minimizing $V_{s\approx}$, the model is incentivized to push values away from the middle range. This enhances the contrast between important and unimportant features. 2) Improving Discriminability: By penalizing values that fall into the middle range, the model is compelled to make clearer distinctions between features, thereby improving the interpretability of the mask.

The loss function effectively penalizes elements of the mask that fall into undesirable ranges, guiding the model to focus on a sparse set of highly relevant features while clearly distinguishing between important and unimportant ones. This optimization contributes to a more robust feature selection mechanism. The combination of sparsity and discriminability helps reduce overfitting and improves generalization, leading to better predictive performance.

To ensure the accuracy of the predicted structure explanation, we compute the structural loss $\mathcal{L}_S$ using the Binary Cross-Entropy loss [25] to compare the actual edge structure with the learned edge strength:

$$\mathcal{L}_S = -\frac{1}{N_E} \sum_{e=1}^{N_E} \left[ \hat{E}_e \log m_e + (1 - \hat{E}_e) \log(1 - m_e) \right], \qquad (9)$$

where $\hat{E} \in \mathbb{R}^{N_E}$ is the edge label vector, with $N_E$ representing the total number of sampled edges. Each element $\hat{E}_e$ indicates whether the corresponding edge $m_e$ exists (1 if exists, 0 if absent). This loss guides $M_s$ in learning the true structural distribution.

Similar to the structure mask $M_s$ and ensure sparsity and enforce discriminability, we compute the feature penalty loss $\mathcal{L}_{FP}$ by penalizing elements of the mask that fall into undesirable ranges, which calculated as follows:

$$\mathcal{L}_{FP} = \lambda_{L1} \cdot \sum_{m \in M_f} |m| + \left| (\delta_{avg}^2 \cdot d_{norm}) - \sum_{i=1}^{N} \left( V_{f>}^i + V_{f<}^i - V_{f\approx}^i \right) \right|, \qquad (10)$$

where $V_{f>}, V_{f\approx}$ and $V_{f<}$ represent the elements of $M_f$ filtered and penalized based on the threshold, calculated similarly to Equation 7.

**Self-Explainable Joint Training.** The normalized features of the original heterogeneous graph are input into the HGNN, passing through $k$ HGNN layers, followed by L2 regularization and log softmax activation to produce the initial predictions $Y_P \in \mathbb{R}^{N_p \times 1}$, $N_p$ is the number of nodes to predict. Similarly, the normalized node features and edges, after being multiplied by the feature mask $M_f$ and the structural mask $M_s$, respectively, are processed through the same $k$ HGNN layers, L2 regularization, and log softmax activation to generate the explanation-enhanced predictions $Y_F \in \mathbb{R}^{N_p \times 1}$.

To compute the loss between the explanation-enhanced predictions $Y_F$ and ground truth labels $Y$ using the Binary Cross-Entropy loss, we employ the following formal definition:

$$\mathcal{L}_F = -\frac{1}{N_p} \sum_{i=1}^{N_p} \left[ Y_i \log(Y_{F,i}) + (1 - Y_i) \log(1 - Y_{F,i}) \right], \qquad (11)$$

where $Y_{P,i}$ represents the predicted probability of the $i$-th sample belonging to the positive class, $Y_i$ is the label for the $i$-th sample. And the loss between the predicted labels $Y_{Pre}$ and the ground truth labels $Y$ using the Negative Log-Likelihood loss [37], we define the following formula:

$$\mathcal{L}_P = -\frac{1}{N_p} \sum_{i=1}^{N_p} \log P(Y_i | Y_{P,i}), \qquad (12)$$

Here, $P(Y_i | Y_{P,i})$ is the probability assigned by the model to the correct label $Y_i$ under the prediction $Y_{P,i}$.

**Training Objectives.** The training objectives for optimizing the overall explanation training loss $\mathcal{L}_{ET}$ are defined as follows:

$$\mathcal{L}_{ET} = \alpha \left[ \beta(\mathcal{L}_{SP} + \mathcal{L}_{FP}) + (1 - \beta)(\mathcal{L}_S + \mathcal{L}_F) \right] + (1 - \alpha)\mathcal{L}_P, \quad (13)$$

where $\alpha$ is used to balance accuracy and explainability and $\beta$ is used to balance penalty and explanation losses. $\beta$ is set to around $0.5\alpha$ in practice. These combined objectives work together to optimize the model's ability to generate accurate, interpretable explanations while preserving strong predictive capabilities.

## 4.2 Enhanced Contrastive Learning

To better align the insights explanations with predictive performance, enhanced contrastive learning leverages the explanations generated during the Self-Explainable training stage. By sampling key nodes from these explanations, contrastive learning is employed to enhance the representation of heterogeneous graph networks.

**Explanatory Graph Node Sampling.** We sample negative and positive nodes around each target node and apply contrastive learning to train the node representation in an unsupervised method. For each node $v_i$, we select $N^+$ sampled positive nodes from its neighboring nodes $N_{v_i}$ that share the same type $r \leftarrow \mathcal{T}_v(v_i)$ sampled from the masked explanatory graph $G_E$, and $N^-$ negative nodes that differ from its type. After passing the explanatory graph through $k$ HGNN layers, we obtain the explanation output features $\hat{H}_Q \in \mathbb{R}^{N \times C}$. We construct positive and negative sample representations $\hat{H}_P \in \mathbb{R}^{N \times N^+ \times C}$ and $\hat{H}_N \in \mathbb{R}^{N \times N^- \times C}$ for each node based on the previously selected samples using these features. Inspired by SimCLR [5], to reduce noise in the representations and facilitate more effective downstream contrastive loss learning, we apply an MLP layer to process the node representations. This yields the refined representations $H_Q, H_P$, and $H_N$, with their dimensions preserved.

**Explanation Enhanced Contrastive Learning.** Here, we train a new HGNN using the explanatory graph. The network architecture consists of $k$ heterogeneous convolution layers, followed by L2 normalization and Log SoftMax activation, yielding the final prediction results $Y_{ET}$. We calculate the loss between the output from the explanatory graph and the ground truth labels:

$$\mathcal{L}_{EP} = -\frac{1}{N} \sum_{i=1}^{N} \log P(Y_i | Y_{ET,i}), \qquad (14)$$

To enable HGNN to learn more nuanced heterogeneous explanatory features, we introduce a contrastive learning mechanism. Specifically, we use the Normalized Temperature-scaled Binary Cross-Entropy loss [5] to calculate the distinguished representation between negative and positive node representations, which integrates elements of binary cross-entropy with temperature scaling to effectively manage contrastive learning objectives, particularly in multi-classification tasks.

$$l_{ij} = -y_{ij} \log \sigma \left( \frac{sim(h_i, h_j)}{\tau} \right) - (1 - y_{ij}) \log \sigma \left( \frac{1 - sim(h_i, h_j)}{\tau} \right), \qquad (15)$$

$$\mathcal{L}_{Con} = \sum_{i=1}^{N} \left( \frac{1}{N^+} \Sigma_{j=1}^{N^+} 1_{ij}^+ l_{ij} + \frac{1}{N^-} \Sigma_{j=1}^{N^-} 1_{ij}^- l_{ij} \right), \qquad (16)$$

where $sim(h_i, h_j)$ denotes the similarity score (cosine) between the representation vectors $h_i$ and $h_j$, with $h_j$ being either a positive sample $h_i^+$ or a negative sample $h_i^-$, $y_{ij} = 1$ when $h_j = h_i^+$ (positive sample and $y_{ij} = 0$ when $h_j = h_i^-$ (negative sample). The notation $1_{ij}^+$ and $1_{ij}^-$ is an indicator function that takes the value of 1 if the condition inside the subscript is true, and 0 otherwise. $\tau$ is the temperature hyperparameter that affects the similarity scales and helps smooth similarity distributions.

**Training Objectives.** The training objectives for optimizing the overall contrastive training loss $\mathcal{L}_{CT}$ are defined as follows:

$$\mathcal{L}_{CT} = \gamma \mathcal{L}_{EP} + (1 - \gamma)\mathcal{L}_{Con}, \qquad (17)$$

the parameter $\gamma$ is employed to balance the two losses. The prediction can be further refined through those insight explanations and contrastive learning-enhanced representations, leading to more accurate and robust outcomes.

## 5 EXPERIMENT

### 5.1 Heterogeneous Benchmark Graphs

To evaluate the performance of the Self-Explainable Heterogeneous Graph Neural Network (SEHG) on heterogeneous node classification tasks, we utilize four representative benchmark datasets: DBLP from [24], IMDB curated by [24], ACM from HAN [34], and Freebase[3]. Detailed information about these datasets can be found in AppendixB.1. Dataset statistics are summarized in Table 1.

| Dataset | Nodes | Nodes Types | Edges | Edges Types | Target | Classes |
|---------|-------|-------|--------|-------|--------|---------|
| DBLP | 26,128 | 4 | 239,566 | 6 | author | 4 |
| IMDB | 21,420 | 4 | 86,642 | 6 | movie | 5 |
| ACM | 10,942 | 4 | 547,872 | 8 | paper | 3 |
| Freebase | 180,098 | 8 | 1,057,688 | 36 | book | 7 |

**Table 1: Dataset statistics of benchmark graphs.**

### 5.2 Baselines

We compare the prediction performance of SEHG on the node classification task with the following strong baselines, which include both representative homogeneous graph neural networks: **GCN** [15], **GAT** [29], and heterogeneous graph neural networks: **RGCN** [28], **HAN** [34], **SHGCN** [24], **SHGAT** [24], **SeHGNN** [36], **HetGNN** [42], **HGT** [10]. For heterogeneous graph explanation tasks, we compare performance using the following robust methods: **GRAD**, **ATT** [29], **GNNExplainer** [38], **PGMExplainer** [30], **GraphLIME** [12], **ProtGNN** [43], and **SES** [13]. The detailed information of all baselines is shown in Appendix B.2.

### 5.3 Predicted Performance Evaluation

The node representations generated by different models are used to predict the node labels in the test set, with performance evaluated using Macro-F1 and Micro-F1 scores. A comprehensive performance comparison is presented in Table 2.

Notably, SEHG (SHGAT) consistently outperforms all other methods across the datasets. On DBLP, SEHG achieves 95.53% (Macro-F1) and 96.15% (Micro-F1), surpassing the second-best SeHGNN by approximately 1.44% and 1.81%, respectively. Similarly, SEHG shows significant improvements over the second-best SHGAT on

IMDB, with increases of 3.91% and 2.31% in Macro-F1 and Micro-F1 scores. This trend continues across ACM and Freebase, where SEHG delivers the highest scores. Moreover, SEHG also demonstrates considerable improvements across different backbones. Notably, on the DBLP dataset, SEHG (HAN) outperforms HAN by 1.34%, while SEHG (SHGCN) shows a 1.04% gain over SHGCN. On IMDB, SEHG achieves larger improvements, with SEHG (HAN) surpassing HAN by 5.48% and SEHG (SHGCN) outperforming SHGCN by 2.53%. For the ACM dataset, SEHG maintains competitive performance with gains of up to 2.28%. On the Freebase dataset, SEHG achieves its most significant improvements, with SEHG (HAN) and SEHG (SHGCN) boosting performance by 13.90% and 15.69%, respectively, demonstrating the effectiveness and robust performance of the proposed heterogeneous explanation framework.

Homogeneous methods, such as GCN, GAT, and RGCN generally underperform compared to heterogeneous graph prediction models. Besides the proposed SEHG, the heterogeneous models SHGAT and SeHGNN also exhibit strong predictive performance, as they are specifically designed to enhance prediction accuracy. Under the other representative backbones GCN and HAN, SEHG improved the performance of backbone graph neural networks by large margins on IMDB and Freebase, which demonstrates that even with relatively weak backbones, SEHG maintains robust prediction performance.

### 5.4 Explanation Qualification

To assess the quality of the explanations in real-world datasets, we rank the features generated by each method based on their importance and remove the top 5 and 10 most significant features to evaluate the impact on predictive performance. For consistency, we utilize SHGAT as the predictive model and employ fidelity $F_{acc}^+$ [1] as the evaluation metric, which can be computed using the following formula:

$$F_{acc}^+ = \frac{1}{N} \sum_{i=1}^{N} (\mathbb{1}(\hat{y}_i = y_i) - \mathbb{1}(\hat{y}_i^{1-m_i} = y_i)), \qquad (18)$$

where $y_i$ represents the original prediction, and $1 - m_i$ denotes the complementary mask that eliminates the crucial features. The indicator function $\mathbb{1}(\hat{y}_i = y_i)$ evaluates to 1 when $y_i$ and $\hat{y}_i$ are equal, and 0 otherwise. The fidelity $F_{acc}^+$ calculates accuracy changes after removing crucial features and structures according to explanations of models. So a higher fidelity value indicates a greater change in model performance, suggesting that the important features identified in explanations are indeed crucial for making accurate predictions. The results are presented in Table 3. The xPath is based on metapath disturbances and is not suitable for this task.

Notably, SEHG consistently achieves the highest fidelity scores across all datasets, demonstrating its superior ability to identify and leverage important features. For example, on the DBLP dataset, SEHG achieves a top-10 fidelity of 85.34%, far surpassing the next-best model, SES by 46.57%. Similar trends are observed on IMDB, ACM, and Freebase, with SEHG outperforming competing models by substantial margins. Specifically, SEHG achieves a top-10 fidelity of 57.61 on IMDB, 76.61% on ACM, and 41.76% on Freebase, surpassing the second-best model by 24.43%, 36.24%, and 21.65%,

**Table 2: Prediction result evaluation. The highest value in each category is highlighted in bold, and the second-highest value is indicated with an underline.**

| Dataset / Model | DBLP Macro-F1 | DBLP Micro-F1 | IMDB Macro-F1 | IMDB Micro-F1 | ACM Macro-F1 | ACM Micro-F1 | Freebase Macro-F1 | Freebase Micro-F1 |
|---|---|---|---|---|---|---|---|---|
| GCN | 89.88±0.15 | 90.17±0.03 | 55.82±0.09 | 62.47±0.05 | 89.87±0.01 | 90.31±0.14 | 31.38±0.02 | 48.20±0.15 |
| GAT | 91.37±0.07 | 92.01±0.15 | 58.84±0.19 | 63.51±0.15 | 92.33±0.11 | 91.15±0.15 | 40.00±0.08 | 63.33±0.17 |
| RGCN | 91.52±0.17 | 92.07±0.12 | 58.85±0.14 | 62.05±0.01 | 91.55±0.14 | 91.41±0.14 | 46.78±0.19 | 58.33±1.62 |
| HAN | 91.67±0.19 | 92.05±0.01 | 57.74±0.11 | 64.61±0.15 | 90.52±0.13 | 89.86±0.06 | 23.31±0.05 | 52.03±0.05 |
| SHGCN | 90.62±0.27 | 91.43±0.31 | 57.68±1.23 | 64.82±0.43 | 92.13±0.10 | 92.19±0.20 | 30.65±1.96 | 60.23±0.27 |
| SHGAT | 93.82±0.05 | 94.22±0.15 | 64.50±0.02 | 67.46±0.05 | 93.17±0.06 | 92.07±0.15 | 44.68±0.02 | 65.63±0.08 |
| SeHGNN | 94.09±0.05 | 94.34±0.01 | 65.09±0.12 | 66.17±0.12 | 91.09±0.14 | 93.44±0.19 | 48.33±0.17 | 59.12±0.01 |
| HGT | 88.70±0.19 | 89.01±0.14 | 61.50±0.12 | 65.88±0.04 | 93.50±0.08 | 92.30±0.03 | 30.71±0.08 | 61.29±0.12 |
| ProtGNN | 79.77±0.02 | 82.77±0.12 | 40.20±0.16 | 42.30±0.03 | 81.32±0.15 | 86.02±0.09 | 31.93±0.05 | 39.13±0.17 |
| SES | 87.39±0.11 | 89.64±0.17 | 55.93±0.21 | 60.27±0.31 | 86.10±0.09 | 89.64±0.17 | 42.37±0.35 | 55.38±0.13 |
| SEHG (HAN) | 93.01±0.18 | 94.22±0.15 | 63.22±0.20 | 66.35±0.17 | 90.10±0.22 | 92.14±0.33 | 31.21±0.42 | 65.30±0.20 |
| SEHG (SHGCN) | 91.59±0.20 | 92.47±0.11 | 60.21±0.07 | 65.49±0.18 | 93.11±0.10 | 92.73±0.22 | 46.34±0.15 | 64.71±0.19 |
| SEHG (SHGAT) | **95.53±0.19** | **96.15±0.15** | **68.41±0.09** | **69.77±0.12** | **95.74±0.09** | **95.21±0.08** | **51.17±0.18** | **66.23±0.16** |

**Table 3: The fidelity $F_{acc}^{+}$% after removing the top important nodes. The highest value is highlighted in bold and the second-highest value is indicated with an underline.**

| Dataset / Model | DBLP $F_{acc}^{+}$ (Top5) | DBLP $F_{acc}^{+}$ (Top10) | IMDB $F_{acc}^{+}$ (Top5) | IMDB $F_{acc}^{+}$ (Top10) | ACM $F_{acc}^{+}$ (Top5) | ACM $F_{acc}^{+}$ (Top10) | Freebase $F_{acc}^{+}$ (Top5) | Freebase $F_{acc}^{+}$ (Top10) |
|---|---|---|---|---|---|---|---|---|
| GRAD | 8.65±0.22 | 11.23±0.17 | 10.74±0.74 | 13.17±0.66 | 6.65±0.31 | 9.93±0.33 | 4.95±0.17 | 6.28±0.23 |
| ATT | 9.68±0.10 | 15.19±0.21 | 10.68±0.41 | 14.97±0.12 | 10.73±0.15 | 16.19±0.35 | 3.77±0.45 | 5.64±0.16 |
| GraphLIME | 10.11±0.15 | 27.44±0.27 | 15.23±0.33 | 21.37±0.47 | 11.91±0.29 | 25.17±0.41 | 4.27±0.26 | 8.32±0.29 |
| GNNExplainer | 25.37±0.26 | 32.11±0.44 | 18.99±0.38 | 25.89±0.36 | 23.28±0.26 | 26.54±0.43 | 15.34±0.11 | 20.11±0.21 |
| PGMExplainer | 27.31±0.65 | 33.21±0.21 | 20.24±0.29 | 33.18±0.74 | 21.86±0.27 | 32.45±0.18 | 8.87±0.19 | 11.63±0.32 |
| SES | 32.17±0.49 | 38.77±0.21 | 28.79±0.27 | 32.54±0.19 | 35.28±0.48 | 40.37±0.35 | 15.27±0.50 | 19.73±0.34 |
| **SEHG** | **76.31±0.19** | **85.34±0.18** | **46.07±0.25** | **57.61±0.21** | **67.88±0.14** | **76.61±0.31** | **32.27±0.15** | **41.76±0.10** |

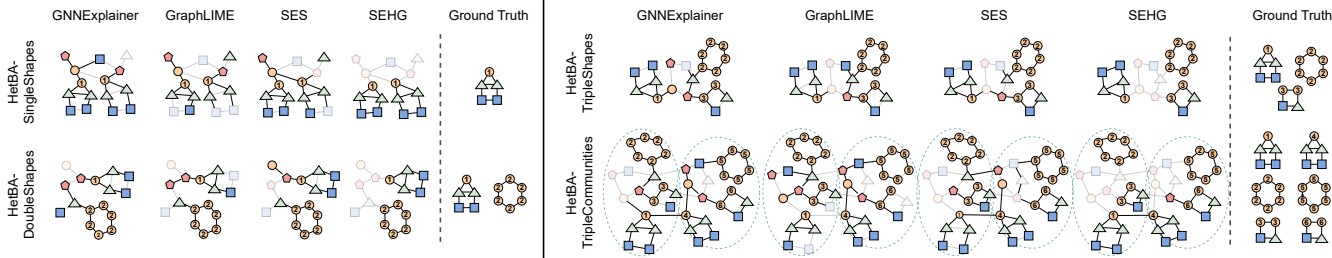

**Figure 3: Evaluation of explanations on HetBA. Example explanation subgraphs for node classification tasks on four synthetic datasets are presented. Each method provides explanations for the orange node's prediction.**

respectively. SES ranks second on the DBLP and ACM datasets. GNNExplainer is securing the second-best position on Freebase. GRAD and ATT models perform comparably and GraphLIME slightly outperforms both. These results highlight SEHG's robustness and effectiveness in extracting critical features across diverse datasets, emphasizing its strong performance in heterogeneous graphs.

To further investigate the explanation quality of SEHG, we evaluate the model using various perturbation-based structural explanation methods, with detailed results provided in Appendix B.4.

## 5.5 Explanation Evaluation on HetBA

The real-world datasets do not have ground truth to verify the explanation's precision. We evaluated the model's explanation accuracy by predictions across four HetBA subsets: SingleShapes, DoubleShapes, TripleShapes, and TripleCommunities. Example results are presented in Fig. 3. Each node belongs to a distinct heterogeneous type (denoted by different colors), to generate an explanation for a given prediction (highlighted in orange) by identifying the subgraph that led to the correct prediction, as specified by the ground

truth. As the number of shapes increases, generating accurate explanations becomes more challenging.

Notably, SEHG exhibits a superior capacity to capture ground truth structures in most cases accurately. On the SingleShapes, DoubleShapes, and TripleShapes datasets, SEHG is the only method that fully identifies the correct ground truth structures, significantly outperforming other methods such as GNNExplainer, GraphLIME, and SES. The compared methods frequently either misidentify irrelevant neighboring nodes or fail to capture essential parts of the explanation subgraph. These inaccuracies emphasize the limitations of using homogeneous methods in heterogeneous graph tasks and underscore the effectiveness of SEHG's approach in addressing these challenges.

We also provide detailed information about the dataset and each method's complete quantitative validation results, as presented in Appendix A.2.

## 5.6 Case Study

In this case study, we examine the explanation results of SEHG and GNNExplainer on the DBLP dataset, as shown in the figure. To investigate the key factors influencing the classification of a specific author (author-428), we analyzed three papers authored by this individual, along with the associated terms and conference information. The thickness of the edges in the figure represents the strength of the influence. SEHG's explanation reveals that paper-6207 and paper-6089 are critical for the classification of author-428, as both papers belong to term-4242 and were presented at conference-9. In contrast, paper-9886 shows some association with author-428 but contributes less significantly to the prediction. On the other hand, GNNExplainer provides less distinction in edge thickness, indicating minimal differentiation in the contribution levels of the various factors. This experiment demonstrates that SEHG offers more insightful explanations with clearer distinctions in the factors contributing to the prediction.

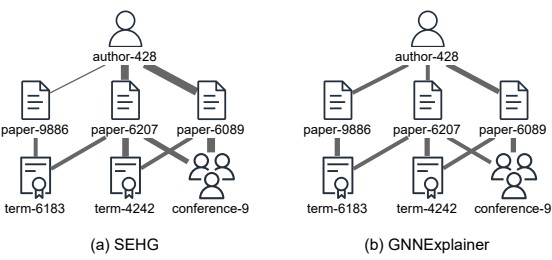

(a) SEHG    (b) GNNExplainer

**Figure 4: Explanation case for SEHG and GNNExplainer on author-428 in DBLP.**

To investigate how different threshold $\delta_{\text{avg}}$ in the penalty loss affect sparsity, we conducted a case study, as shown in Appendix B.5.

## 5.7 Ablation Studies

The ablation study presented in Table 4 highlights the impact of SEHG's components on prediction and explanation performance, including feature masks $M_f$, structure masks $M_s$, feature penalty loss $\mathcal{L}_{\text{FP}}$, structure penalty loss $\mathcal{L}_{\text{SP}}$, and SimCLR. Removing any component leads to performance declines, with the most significant

**Table 4: Ablation studies evaluating the impact of $M_f$, $M_s$, $\mathcal{L}_{\text{FP}}$, $\mathcal{L}_{\text{SP}}$, and SimCLR (SC). The $\mathcal{L}_{*\text{P}}$, refers to $\mathcal{L}_{\text{FP}}$ + $\mathcal{L}_{\text{SP}}$. $M_{\text{GEX}}$ and $M_{\text{PGM}}$ denote the masks generated by GNNExplainer and PGMExplainer, respectively.**

| Dataset | DBLP | | IMDB | |
|---|---|---|---|---|
| Model | Micro-F1 | $F_{acc}^+$(Top10) | Micro-F1 | $F_{acc}^+$(Top10) |
| SEHG | **96.15±0.15** | **85.34±0.18** | **69.77±0.12** | **57.61±0.21** |
| SEHG - $M_f$ | 90.26±0.11 | 60.33±0.24 | 61.34±0.12 | 41.65±0.19 |
| SEHG - $M_s$ | 93.24±0.13 | 72.27±0.19 | 65.31±0.17 | 49.27±0.24 |
| SEHG - $\mathcal{L}_{\text{FP}}$ | 89.64±0.19 | 55.31±0.16 | 58.28±0.23 | 46.29±0.39 |
| SEHG - $\mathcal{L}_{\text{SP}}$ | 93.15±0.16 | 66.21±0.15 | 62.81±0.25 | 47.20±0.28 |
| SEHG - SC | 87.50±0.19 | 57.00±0.11 | 57.35±0.15 | 42.38±0.20 |
| SEHG - ($\mathcal{L}_{*\text{P}}$+SC) | 85.43±0.30 | 29.01±0.21 | 55.32±0.23 | 16.38±0.37 |
| SEHG + $M_{\text{GEX}}$ | 94.26±0.19 | – | 68.83±0.24 | – |
| SEHG + $M_{\text{PGM}}$ | 95.14±0.24 | – | 68.26±0.20 | – |

drop in prediction (Micro-F1) observed when SimCLR is removed (8.65% on DBLP and 12.42% on IMDB), demonstrating its key role in learning node representations. Feature-related components have a larger impact than structure-related ones. For example, removing $M_f$ and $\mathcal{L}_{\text{FP}}$ leads to a greater drop in explanation performance (up to 30.03% on DBLP) than removing $M_s$ and $\mathcal{L}_{\text{SP}}$. When both penalty losses and SimCLR are removed, SEHG suffers a 56.33% decline in explanation fidelity on DBLP, emphasizing their critical importance. We also explored replacing $M_s$ with the masks from GNNExplainer $M_{\text{GEX}}$ and PGMExplainer $M_{\text{PGM}}$ during SEHG's second-stage contrastive learning phase. This resulted in slight performance drops compared to SEHG, demonstrating the effectiveness of SEHG's Self-Explainable mask generation mechanism. The minor variations further highlight the robustness of the contrastive learning framework.

We also investigated the importance of Penalty Loss and SimCLR when applied to different backbones and more datasets, as detailed in Appendix B.6.

## 6 CONCLUTION

In this paper, we present SEHG, an innovative self-explainable heterogeneous graph neural network framework that integrates explainability into the learning process of HGNNs. SEHG utilizes a unique heterogeneous graph explanation generator that produces node feature and edge structure masks. These masks are incorporated into the original graph, guiding the model to focus on the most crucial aspects of the data, thereby promoting more accurate predictions and explanations. In addition, we introduced a group of synthetic datasets, HetBA, and evaluated the performance of different explanation models. Our extensive experiments on both real-world and synthetic datasets further validated the effectiveness of SEHG. However, challenges persist in developing self-explainable models for larger or dynamic heterogeneous graphs, which warrant further investigation in future studies.

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

# A  HetBA DATASET

## A.1  Dataset Introduction

Inspired by GNNExplainer [38] and utilizing the code provided by [16], we constructed four heterogeneous datasets to quantitatively evaluate explainability performance and enhance the visual assessment of explanations.

**HetBA-SingleShapes:** This dataset consists of 19,500 nodes. Of these, 18,000 nodes are generated based on a base Barabási-Albert (BA) graph. An additional 300 five-node "house"-structured motifs are randomly attached to nodes in the BA graph, followed by perturbation with $0.1 \times N$ random edges. As illustrated in Fig. 3, each node in the BA graph is randomly assigned a type (e.g., triangle, square, circle, or pentagon), while nodes in the house motif are assigned the types of circle, triangle, and square for the top, middle, and bottom nodes, respectively. Nodes of the circle type are given labels for classification, with all top-positioned circle nodes in the house motif sharing the same label. In total, there are two node labels. The node features are extracted from a diagonal matrix composed of nodes of the same type. For edges, we assign 16 distinct labels based on the source and target node types.

**HetBA-DoubleShapes:** This dataset contains 39,600 nodes. The generation process is similar to HetBA-SingleShapes, but the number of BA graph nodes is increased to 36,000, with 600 five-node house motifs and 100 six-node cycle motifs added sequentially. The node labeling in house motifs remains the same as described earlier, while each node in the cycle motifs is assigned a unique new label, resulting in 3 total node labels. Since no new node types are introduced, the dataset maintains the same 16 edge labels.

**HetBA-TripleShapes:** Following a process similar to HetBA-DoubleShapes, this dataset contains 61,200 nodes. It consists of 54,000 BA graph nodes, 900 house motifs, 150 cycle motifs, and 450 "family of four" motifs. In the "family of four" motif, the parents are represented by square and triangle types, while the children are circles, each assigned a new unique label. The node and edge labels follow the same logic as the earlier datasets, resulting in 4 node labels and 16 edge labels.

**HetBA-TripleCommunities:** This dataset is constructed from two HetBA-TripleShapes graphs, resulting in a total of 122,400 nodes. Labels are assigned based on the motif's origin within the two HetBA-TripleShapes graphs, leading to 7 node labels and 16 edge labels.

## A.2  Quantitative Evaluation on HetBA

We validated the performance of SEHG and comparative methods on four subsets of the HetBA dataset. The Table 5 presents a comparative evaluation of explanation performance across various models on the HetBA dataset. Key metrics include Accuracy and Micro-F1 scores across four distinct datasets: SingleShapes, DoubleShapes, TripleShapes, and TripleCommunities.

The SEHG model consistently outperforms other models across all datasets. Specifically, SEHG achieves the highest Accuracy and Micro-F1 scores in each category, underscoring its superior performance in explanation tasks. For example, in the SingleShapes dataset, SEHG attains an Accuracy of 90.68 and a Micro-F1 score of 92.33, significantly surpassing other models. This trend is evident across all datasets, with SEHG leading in both metrics. Notably,

SEHG demonstrates remarkable superiority in Accuracy, with a 9.11% improvement over the second-best model, GNNExplainer, on the DoubleShapes dataset. Additionally, on the TripleCommunities dataset, SEHG outperforms the second-highest model by 9.44%. These substantial margins highlight SEHG's exceptional effectiveness in providing accurate and reliable explanations.

The results demonstrate SEHG's exceptional ability to provide accurate and reliable explanations. This consistent superiority underscores SEHG's effectiveness in heterogeneous explanation tasks, offering a clear advantage in both explainability and accuracy.

# B  EXPERIMENT DETAIL

## B.1  Datasets Details

We present the details of four real-world heterogeneous datasets:

**DBLP** is a prominent bibliographic database for computer science research. We use a subset from [24], which includes nodes representing authors, papers, terms, and venues across four domains.

**IMDB** is a movie database where we use a subset from the Action, Comedy, Drama, Romance, and Thriller categories, as curated by [24].

**ACM** is also a citation network, with a subset from HAN [34], retaining all paper citations and references.

**Freebase** [3] is a large knowledge graph, from which we sample a subgraph featuring 8 entity types and approximately 1,000,000 edges, following the methodology in a prior survey.

## B.2  Baselines Details

The detailed information on baselines is shown here:

**GCN** [15] is a classic graph convolution network proposed by simplifying the process of ChebNet [7].

**GAT** [29] is a graph attention network that applies a self-attention mechanism to calculate the weights of neighboring nodes.

**RGCN** [28] is a relational graph convolution network using relation-specific weights and optimizing the model through weight sharing and sparsity constraints.

**HAN** [34] is an early heterogeneous attention network that uses multiple manually defined meta-paths and applies hierarchical attention to learn node and semantic weights.

**SHGCN** and **SHGAT** [24] are straightforward HGNs based on GCN and GAT architectures, respectively, complemented by linear layers, which are carefully tuned to achieve robust performance.

**SeHGNN** [36] uses a lightweight mean aggregator for neighbor pre-computation, a single-layer structure with extended meta-paths for an enlarged receptive field, and a transformer-based method for feature integration from various meta-paths.

**HetGNN** [42] Introduce a random walk with a restart to sample a fixed number of strongly correlated heterogeneous neighbors per node, grouping them by type, followed by a two-module neural network to aggregate feature information from these sampled neighbors.

**HGT** [10] Heterogeneous graph transformer is a transformer-based model by subgraph sampling strategies to accelerate heterogeneous graph computations for support web-scale graphs.

 

**Table 5: Explanation performance evaluation on HetBA. The highest value in each category is highlighted in bold, and the second-highest value is indicated with an underline.**

| Dataset Model | SingleShapes | | DoubleShapes | | TripleShapes | | TripleCommunities | |
|---|---|---|---|---|---|---|---|---|
| | Accuracy | Micro-F1 | Accuracy | Micro-F1 | Accuracy | Micro-F1 | Accuracy | Micro-F1 |
| GRAD | 72.85±0.48 | 74.98±0.36 | 68.29±0.42 | 69.35±0.29 | 67.55±0.22 | 66.89±0.41 | 62.78±0.48 | 62.34±0.33 |
| ATT | 73.12±0.35 | 75.41±0.28 | 69.01±0.41 | 70.12±0.34 | 66.98±0.36 | 67.51±0.37 | 64.01±0.38 | 63.11±0.31 |
| GraphLIME | 81.23±0.22 | 82.17±0.20 | 78.31±0.26 | 79.24±0.18 | 76.45±0.24 | 77.01±0.25 | 72.99±0.32 | 71.75±0.28 |
| GNNExplainer | 82.01±0.28 | 83.12±0.25 | 79.14±0.30 | 79.99±0.22 | 77.25±0.23 | 77.44±0.19 | 73.53±0.27 | 72.21±0.30 |
| PGMExplainer | 81.12±0.31 | 86.33±0.22 | 78.45±0.27 | 79.34±0.23 | 76.01±0.21 | 76.14±0.26 | 72.41±0.35 | 71.22±0.29 |
| SES | 85.19±0.17 | 84.05±0.18 | 75.22±0.19 | 81.11±0.12 | 81.05±0.22 | 81.47±0.24 | 71.44±0.31 | 73.36±0.22 |
| **SEHG** | **90.68±0.23** | **92.33±0.12** | **88.25±0.13** | **88.94±0.16** | **86.15±0.19** | **86.38±0.21** | **82.97±0.32** | **81.71±0.26** |

**Table 6: Impact of different perturbation methods on model explanations and their influence on prediction results. Lower values indicate greater result fluctuations, suggesting more accurate explanations. Optimal values are highlighted in bold.**

| Perturbation Method | Metric | DBLP | | | | IMDB | | | | ACM | | | |
|---|---|---|---|---|---|---|---|---|---|---|---|---|---|
| | | GNNE | HGT | SHGAT | **SEHG** | GNNE | HGT | SHGAT | **SEHG** | GNNE | HGT | SHGAT | **SEHG** |
| PGM-Explainer | $F_{acc}$ | 7.9 | 6.2 | 31.9 | **2.7** | 9.8 | **3.4** | 9.4 | 7.9 | 32.9 | 46.7 | 31.7 | **27.9** |
| | $F_{prob}$ | 13.8 | 5.9 | 11.3 | **1.1** | 7.2 | 4.8 | 2.0 | **1.2** | 17.1 | 35.2 | **11.4** | 15.5 |
| GEM | $F_{acc}$ | 35.2 | **6.1** | 32.1 | 12.4 | 22.7 | 13.5 | **17.1** | 18.4 | 37.2 | 1.9 | 26.9 | **1.1** |
| | $F_{prob}$ | 12.3 | 6.3 | 11.4 | **2.1** | 22.7 | 13.3 | 5.1 | **3.2** | 23.8 | 1.6 | 11.3 | **1.2** |
| SubgraphX | $F_{acc}$ | 11.1 | 0.8 | 5.9 | **0.2** | 7.1 | 1.4 | 2.8 | **1.1** | 10.1 | 1.4 | 4.6 | **1.3** |
| | $F_{prob}$ | 7.2 | -2.3 | 2.1 | **-2.8** | 2.3 | 0.1 | -1.6 | **-2.9** | 9.0 | 1.2 | 1.5 | **1.1** |
| xPath | $F_{acc}$ | 4.3 | 0.3 | 0.3 | **0.1** | 5.8 | 0.4 | 2.5 | **0.3** | 8.5 | 5.5 | 4.1 | **0.17** |
| | $F_{prob}$ | 2.2 | -1.0 | -0.4 | **-1.3** | 0.0 | -2.8 | -2.0 | **-3.1** | 13.2 | 1.9 | 0.7 | **-0.2** |

**GRAD** is a gradient-based method. We compute the gradient of the GNN's loss function to the adjacency matrix and the associated node features, similar to a saliency map approach.

**ATT** is a graph attention GNN (GAT) [29] that learns attention weights for edges in the computation graph, which we use as a proxy measure of edge importance.

**GNNExplainer** [38] generate subgraph explanations by maximizing the mutual information between GNN predictions and the distribution of possible subgraph structures.

**PGMExplainer** [30] leverages a probabilistic graphical model to identify essential components, generating an explanation by constructing a PGM that closely approximates the original prediction.

**GraphLIME** [12] Extend the LIME algorithm to deep graph models and investigate the significance of different node features for node classification tasks.

**ProtGNN** [43] combines prototype learning with GNNs and provides a new perspective on the explanations of GNNs.

**SES** [13] is a self-explainable and self-supervised homogeneous graph network by training a feature and structure mask.

## B.3 Experiment Settings

Our experiments comprise two main tasks: prediction tasks on four real-world datasets and explainability tasks on four synthetic datasets. For both tasks, we follow HGB [24] benchmark protocol.

Specifically, edges are visible during training, and node labels are split into 24% for training, 6% for validation, and 70% for testing. For the Freebase dataset used in the prediction tasks and the HetBA-TripleCommunities dataset used in the explainability tasks, SEHG employs the Adam optimizer with a learning rate of 0.002 and no weight decay. In all other experiments, the learning rate is set to 0.001 with a weight decay of 0.005. In the prediction tasks, the Penalty Loss in SEHG uses $\delta_{avg}$ set to 0.35, while for explainability tasks, $\delta_{avg}$ is set to 0.25.

## B.4 Explanation Quality Evaluation Supplement

We adopt the evaluation methodology outlined in previous work [40] to compare the significant explanations generated by various methods (x-axis). The impact on performance after applying four perturbation techniques—PGMExplainer [30], GEM [21], SubgraphX [41], and xPath [20]—is assessed. The results are quantified in terms of $F_{acc}$, which measures the prediction change, and $F_{prob}$, which evaluates the probability change of the original predicted label. These metrics are defined as follows:

$$F_{acc} = \frac{1}{|\mathcal{T}|} \sum_{(v_t,y) \in \mathcal{T}} (1 - 1_{y=\tilde{y}}),$$

$$F_{prob} = \frac{1}{|\mathcal{T}|} \sum_{(v_t,y) \in \mathcal{T}} (M_G(v_t)[y] - M_{\tilde{G}}(v_t)[y]). \tag{19}$$

**Table 7: Ablation study evaluating the effects of Penalty Loss ($\mathcal{L}_{*P}$) and SimCLR (SC) in SEHG across various backbones.**

| Dataset / Model | DBLP Micro-F1 | DBLP $F_{acc}^+$(Top10) | IMDB Micro-F1 | IMDB $F_{acc}^+$(Top10) | ACM Micro-F1 | ACM $F_{acc}^+$(Top10) | Freebase Micro-F1 | Freebase $F_{acc}^+$(Top10) |
|---|---|---|---|---|---|---|---|---|
| SEHG (SHGAT) | **96.15±0.15** | **85.34±0.18** | **69.77±0.12** | **57.61±0.21** | **95.21±0.08** | **76.61±0.31** | **66.23±0.16** | **41.76±0.10** |
| SEHG (SHGAT) - $\mathcal{L}_{*P}$ | 91.20±0.28 | 40.20±0.24 | 60.26±0.19 | 20.91±0.24 | 88.47±0.22 | 18.52±0.30 | 60.77±0.28 | 20.58±0.24 |
| SEHG (SHGAT) - SC | 87.50±0.19 | 57.00±0.11 | 57.35±0.15 | 42.38±0.20 | 85.26±0.25 | 53.76±0.28 | 58.80±0.30 | 45.11±0.22 |
| SEHG (SHGAT) - ($\mathcal{L}_{*P}$+SC) | 85.43±0.30 | 29.01±0.21 | 55.32±0.23 | 16.38±0.37 | 83.79±0.31 | 27.15±0.32 | 55.92±0.29 | 23.21±0.30 |
| SEHG (SHGCN) | **92.47±0.11** | **70.10±0.25** | **65.49±0.18** | **49.93±0.30** | **92.73±0.22** | **58.35±0.28** | **64.71±0.19** | **40.30±0.26** |
| SEHG (SHGCN) - $\mathcal{L}_{*P}$ | 87.39±0.21 | 25.50±0.18 | 58.03±0.25 | 26.20±0.22 | 88.35±0.30 | 25.59±0.20 | 62.27±0.25 | 23.15±0.22 |
| SEHG (SHGCN) - SC | 89.43±0.22 | 39.70±0.19 | 56.19±0.20 | 43.15±0.25 | 85.26±0.30 | 55.18±0.27 | 56.35±0.24 | 40.21±0.28 |
| SEHG (SHGCN) - ($\mathcal{L}_{*P}$+SC) | 85.85±0.25 | 19.40±0.20 | 55.14±0.22 | 15.60±0.18 | 83.79±0.30 | 19.31±0.15 | 52.25±0.29 | 20.12±0.20 |
| SEHG (HAN) | **92.01±0.18** | **59.10±0.23** | **63.22±0.20** | **44.37±0.28** | **90.10±0.22** | **63.04±0.25** | **65.30±0.20** | **40.15±0.26** |
| SEHG (HAN) - $\mathcal{L}_{*P}$ | 90.30±0.20 | 23.70±0.22 | 60.33±0.25 | 24.93±0.20 | 87.58±0.28 | 35.74±0.18 | 61.50±0.25 | 23.10±0.22 |
| SEHG (HAN) - SC | 86.60±0.22 | 51.30±0.18 | 57.30±0.20 | 39.47±0.22 | 85.95±0.30 | 57.94±0.27 | 57.25±0.28 | 35.10±0.26 |
| SEHG (HAN) - ($\mathcal{L}_{*P}$+SC) | 85.44±0.25 | 18.90±0.22 | 55.29±0.20 | 10.70±0.18 | 83.89±0.30 | 16.66±0.22 | 55.40±0.28 | 19.90±0.20 |

Specifically, let $\mathcal{T}$ denote test samples correctly predicted by the model. For a test sample $(v_t, y)$, we induce a graph $\tilde{G}$ based on all the involved graph objects in its explanation. We then compute the prediction $M_{\tilde{G}}(v_t)$ and the predicted label $\tilde{y}$ based on $\tilde{G}$.

The Table 6 highlights that SEHG consistently achieves the lowest values for both $F_{acc}$ and $F_{prob}$ across multiple datasets and perturbation methods, indicating its explanations are highly accurate. In the DBLP dataset, SEHG exhibits the smallest fluctuation in prediction accuracy and probability under methods like PGM-Explainer, showing that its explanations are closely tied to its prediction process. This pattern is seen across various metrics, with SEHG outperforming models such as GNNE, HGT, and SHGAT in minimizing prediction changes after perturbation.

For instance, in the DBLP dataset, SEHG achieves the lowest $F_{acc}$ (2.7) and $F_{prob}$ (1.1) under the PGM-Explainer method, outperforming all other models by minimizing prediction fluctuations. Similarly, in the IMDB dataset, SEHG consistently delivers superior performance, recording the lowest $F_{acc}$ and $F_{prob}$ values across most cases. In the ACM dataset, SEHG once again surpasses competing models, maintaining the best results across all perturbation methods. These results demonstrate that SEHG's predictions are highly influenced by the explanations it generates, suggesting that its accuracy and robustness are deeply connected to how well it captures the structure of the graph during perturbation. Consequently, the model's lower values across different perturbation methods show that its explanations are key factors in influencing graph prediction outcomes.

## B.5 Case Study Supplement

We conducted a case study to examine the effect of varying the threshold $\delta_{avg}$ on $M_s$, as shown in the results. When $\delta_{avg}$ is low, the values of $M_s$ tend to converge, making differentiation difficult. As $\delta_{avg}$ gradually increases, the penalty loss becomes effective, resulting in a clearer distinction between the deep blue and light blue regions within $M_s$ and a noticeable increase in sparsity. This indicates that Penalty Loss, which enforces sparsity and penalizes

mask elements in undesirable ranges, is essential for maintaining prediction accuracy and model explainability.

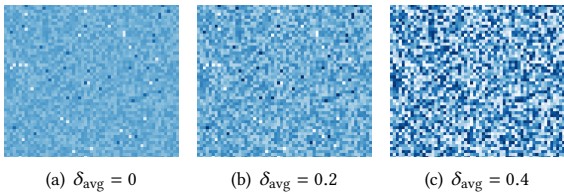

(a) $\delta_{avg} = 0$     (b) $\delta_{avg} = 0.2$     (c) $\delta_{avg} = 0.4$

**Figure 5: Case study in $M_s$ with different $\delta$.**

## B.6 Ablation Studies Supplement

The ablation study provides detailed insights into the effects of Penalty Loss ($\mathcal{L}_{*P}$, refers to $\mathcal{L}_{FP} + \mathcal{L}_{SP}$) and SimCLR (SC) on SEHG's performance across various backbones and datasets, as shown in Table 7. SEHG consistently outperforms its variants, with the SHGAT backbone generally exhibiting the highest performance across all datasets. For instance, in the DBLP dataset, SEHG achieves the top Micro-F1 and $F_{acc}^+$ (Top10) scores, with SHGAT also leading. The removal of either $\mathcal{L}_{*P}$ or SC notably degrades performance, with the largest declines observed in $F_{acc}^+$ (Top10) scores, highlighting the critical role of these components.

The impact of $\mathcal{L}_{*P}$ is particularly significant, as its absence results in substantial reductions in both Micro-F1 and $F_{acc}^+$ (Top10) scores. For example, in the IMDB dataset, removing $\mathcal{L}_{*P}$ leads to a drop of up to 9.50% in Micro-F1 and a dramatic decrease in $F_{acc}^+$ (Top10). This indicates that $\mathcal{L}_{*P}$ plays a crucial role in enhancing the model's performance and accuracy. In contrast, while SimCLR (SC) also contributes positively, its removal results in a less severe drop in performance compared to $\mathcal{L}_{*P}$. The reduction in Micro-F1 scores ranges from 2.67% (DBLP) to 12.42% (IMDB), demonstrating that SC has a significant but slightly less critical impact.

Overall, the study underscores the importance of both $\mathcal{L}_{*P}$ and SC in optimizing SEHG's performance. The consistent advantage

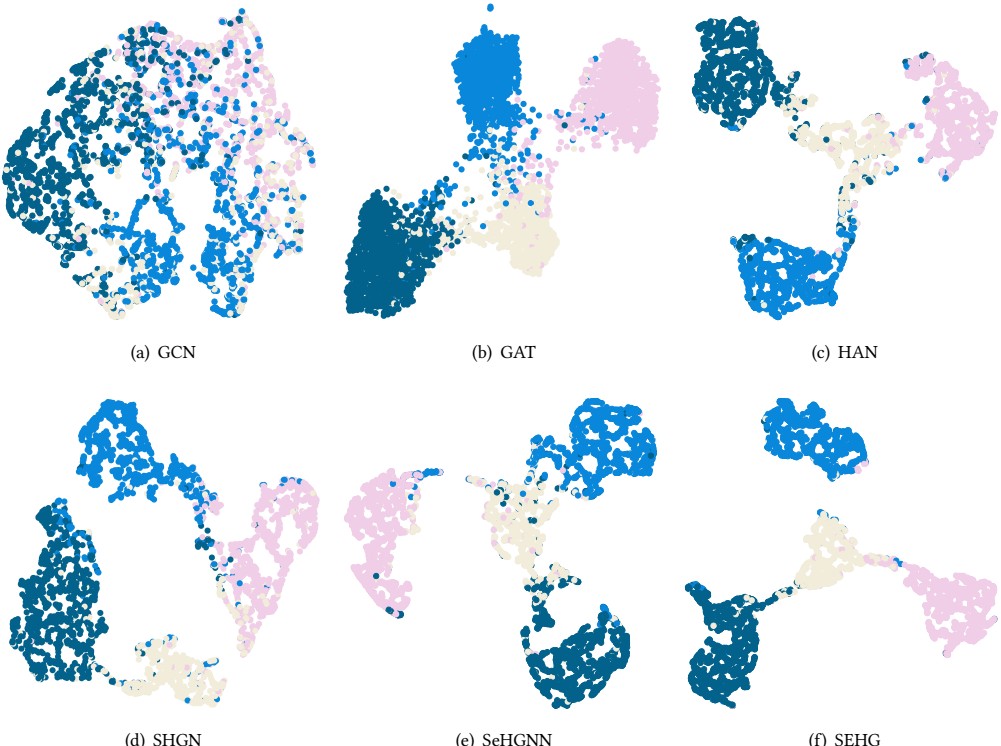

(a) GCN                              (b) GAT                              (c) HAN

(d) SHGN                            (e) SeHGNN                            (f) SEHG

**Figure 6: Visualization of Predictive Quality on DBLP. Different colors represent nodes from different categories.**

of SEHG with different backbones, particularly SHGAT, reflects the effectiveness of these components in improving the model's ability to provide high-quality explanations. The results highlight that while both $\mathcal{L}_{*P}$ and SC are beneficial, Penalty Loss has a more pronounced effect on maintaining and enhancing the model's performance across various datasets.

## B.7 Visualization of Predictive Quality

We utilize t-SNE [17] to visualize the node representations produced by the model on the DBLP dataset, allowing us to explore performance differences between SEHG and baseline variants. Fig. 6

illustrates these visualizations, projecting the node representations generated by SEHG into a two-dimensional subspace.

From Fig. 6, it is evident that the node distribution for GCN is disordered, contrasting with its performance in homogeneous tasks. GAT displays a more structured distribution, with the representations of the four node categories beginning to separate, though some confusion remains in the central region. The results of HAN, SHGAT, and SeHGNN are similar, with the nodes being clearly divided into four categories, but there are still classification errors and insufficient clustering within classes. In contrast, SEHG achieves the most optimal visualization, with nodes clearly separated into four distinct clusters and a high degree of intra-class cohesion, which aligns with SEHG's superior predictive performance.