# OpenReview forum: "SEHG: Bridging Interpretability and Prediction in Self-Explainable Heterogeneous Graph Neural Networks"
_ACM.org/TheWebConf/2025/Conference — WWW 2025 Oral_

### Official Review · Reviewer_c8t7 · 2024-11-17

**Novelty:** 5
**Technical Quality:** 6

**Review:**

The paper presents a new architecture, SEHG, addressing challenges in enhancing the interpretability and performance of heterogeneous graph neural networks (HGNNs), which are widely used in web-based applications involving heterogeneous graph structures. Traditional post-hoc explanation models for HGNNs fail to influence predictions, and self-explainable homogeneous models underperform on heterogeneous graphs. SEHG bridges this gap by integrating explanation generation into the HGNN learning process through two stages: (1) generating explanations alongside predictions, and (2) improving prediction accuracy using contrastive learning. A key idea of SEHG is its learnable heterogeneous masks, which provide structure and feature explanations without relying on manually defined metapaths. These masks are regulated by a range-based penalty to ensure high-quality and sparse explanations. Also, a synthetic heterogeneous dataset collection called HetBA is provided to facilitate quantifying and visualizing explanations for heterogeneous graphs. The experimental results are compelling: SEHG surpasses strong baselines in real-world node classification tasks with performance improvements of up to 3.91%. On synthetic datasets, it achieves results with improvements of up to 9.44% and outperforms competitors in explanation fidelity by up to 46.57%.

The integration of explanation generation directly into the framework, along with the focus on enhancing prediction accuracy, is both novel and highly interesting. While the architecture is complex, it is effectively presented through clear and illustrative figures. It is also commendable that the integration of explanation generation contributes to improved performance, showcasing the practical value of this approach. Moreover, the experimental study is thorough, involving comparisons with numerous baseline methods and datasets, evaluating fidelity by removing nodes identified as important, and including detailed case studies. Overall, I believe this work represents a significant step forward in developing interpretable HGNNs, combining innovative architectural design with robust empirical validation.

Minor comments:
- Explain what a matapath is.
- Figure 2: Sampleing -> Sampling
- For Table 3, it might also be interesting to evaluate the same architecture after removing nodes identified as important by each method to see whether the "importance" is universal or specific to each method.

**Questions:**

No particular question.

**Reviewer Confidence:**

1: The reviewer's evaluation is an educated guess

**Scope:**

3: The work is somewhat relevant to the Web and to the track, and is of narrow interest to a sub-community

---

### Official Review · Reviewer_n9BN · 2024-11-23

**Novelty:** 5
**Technical Quality:** 6

**Review:**

This paper introduces a Self-Explainable Heterogeneous Graph Neural Network (SEHG), a novel approach that integrates explanation generation directly into the learning process of HGNNs. SEHG is capable of producing interpretable structure and feature explanations while employing a contrastive learning strategy to enhance performance. Additionally, this study presents four synthetic heterogeneous datasets designed to assess the quality of HGNN explanations. Experimental results demonstrate that the proposed methods are effective in improving both the performance and interpretability of HGNNs.

### Quality
The paper addresses an important and underexplored area of HGNN research: explanation generation. The proposed methods and datasets have the potential to significantly benefit the research community. However, some quality concerns remain. First, the paper lacks an analysis of computational complexity. Given the method's involvement of multiple modules, a thorough evaluation of space and time complexity is essential. Second, the selection of baseline methods is outdated. Comparing the proposed method with more recent approaches would provide a stronger foundation for evaluating its effectiveness. Detailed feedback on these issues is provided in the Questions part below.

### Clarity
The paper is well-written, clear, and easy to follow. The structure and explanations are accessible to readers.

### Originality
This work introduces a novel method for improving the performance and interpretability of HGNNs, along with new datasets for evaluation. The contributions are original and valuable to the community. However, the lack of publicly available code and datasets for reproduction limits the accessibility and impact of the work.

### Significance
This paper has the potential to significantly advance HGNN research, particularly in terms of explanation generation. However, addressing the noted concerns is critical to maximizing its impact.

**Questions:**

1. Please provide the space and time complexity analysis of the proposed methods. Additionally, an empirical evaluation would greatly enhance the discussion.
2. It is essential to discuss or compare some recent and significant HGNNs, such as HALO [1], HINormer [2], and PSHGCN [3].
3. Publishing the source code and the proposed datasets is crucial for reproducibility and validation.
4. Could the authors share the hyperparameter settings, such as gamma, the number of hidden units per layer, and other relevant details?


[1] Ahn, et al. "Descent steps of a relation-aware energy produce heterogeneous graph neural networks." Advances in Neural Information Processing Systems 35 (2022): 38436-38448.
[2] Mao, et al. "Hinormer: Representation learning on heterogeneous information networks with graph transformer." Proceedings of the ACM Web Conference 2023. 2023.
[3] He, et al. "Spectral Heterogeneous Graph Convolutions via Positive Noncommutative Polynomials." Proceedings of the ACM on Web Conference 2024. 2024.

**Reviewer Confidence:**

4: The reviewer is certain that the evaluation is correct and very familiar with the relevant literature

**Scope:**

4: The work is relevant to the Web and to the track, and is of broad interest to the community

---

### Official Review · Reviewer_bLaz · 2024-11-28

**Novelty:** 5
**Technical Quality:** 5

**Review:**

This paper proposed to solve the interpretability of the HGNN and incorporated with the prediction module to enhance the representation learning.
Pros:
1.	The motivation is clear.
2.	The designed method is targeted.
Cons:
1.	The writing format should be noticed, e.g. the loss LSF is not the same format.
2.	The illustration for Figure 2 should be clearer.

**Questions:**

1.	Is there a method for enhancement using interpretability in homogeneous graphs?  There is no comparison of such methods seen.
2.	Does this enhancement method also have an improvement effect in homogeneous graphs? Compared with the methods of homogeneous graphs, are there any other challenges besides the difference in message-passing?

**Reviewer Confidence:**

4: The reviewer is certain that the evaluation is correct and very familiar with the relevant literature

**Scope:**

4: The work is relevant to the Web and to the track, and is of broad interest to the community

---

### Official Review · Reviewer_Y7bd · 2024-12-02

**Novelty:** 4
**Technical Quality:** 3

**Review:**

The paper proposes a Self-Explainable Heterogeneous Graph Neural Network (SEHG) designed to address the interpretability challenges of Heterogeneous Graph Neural Networks (HGNNs). SEHG integrates explanation generation directly into the model’s training, improving both interpretability and predictive performance. The framework consists of two stages: Heterogeneous Self-Explainable Training and Enhanced Contrastive Learning. The first stage employs learnable masks for node and edge features, guided by a range-based penalty to ensure sparsity and clarity in explanations. The second stage enhances predictions by leveraging explanatory graphs through contrastive learning inspired by SimCLR. To evaluate its effectiveness, the authors introduce a new synthetic dataset, HetBA, designed for heterogeneous graph explanation tasks. Experimental results on real-world and synthetic datasets demonstrate that SEHG outperforms existing methods, achieving state-of-the-art results in both prediction accuracy and explanation quality.

SEHG demonstrates several key strengths.

Pros:

1. One notable advantage is its focus on self-explainable HGNNs, which is well-motivated. Unlike post-hoc explanation methods or those designed for homogeneous graphs, SEHG directly integrates explainability into HGNNs, addressing the complexities of heterogeneous data more effectively and aligning with real-world scenarios.

2. Another strength lies in the comprehensive evaluation experiments. The authors assess both predictive performance (e.g., classification accuracy) and explanation quality (e.g., fidelity metrics), providing a holistic view of the model’s capabilities.

3. Lastly, the paper is well-written and easy to understand. It incorporates sufficient theoretical details, including explicit formulations and loss functions, which enhance the credibility of the work.

However, there are still some issues that need to be addressed.

Cons:

1. The first weakness concerns the design and justification of the range-based penalty in the Heterogeneous Self-Explainable Training phase. As claimed by the authors in the Introduction section, this range-based penalty plays a crucial role in generating both node and edge explanations (Eqs. (8) and (10)). However, the second term of this penalty involves minimizing an absolute value, which encourages it to converge towards zero, leading to an optimization target of $\sum(V_{s>}^{i} + V_{s<}^{i} - V_{s\approx}^{i})$ equating to $\delta_{avg}^{2} d_{norm}$. This optimization to a scalar appears unstable and suboptimal. Moreover, the authors do not justify why this scalar is appropriate, nor do the experiments validate the claim of “Encouraging Extremes.” Additionally, since $\delta_{avg}^{2} d_{norm}$ depends on the edge mask itself, rather than being a fixed scalar from dataset statistics, the penalty’s rationale remains questionable, particularly given its impact on backpropagation.

2. The second weakness relates to the lack of clarity in constructing the HetBA dataset. Although this dataset is presented as a core contribution, the main text provides minimal details on its construction, offering only visualization results. Appendix A.1 mentions inspiration from GNNExplainer and the use of code from a prior work [16], but it does not elaborate on the differences between HetBA and the dataset in [16]. This lack of detail challenges the credibility of HetBA’s contribution and poses difficulties for replication and assessment of its uniqueness.

3. The third weakness concerns the joint training of explanation and prediction. While joint training is an core component of SEHG’s design, the authors do not compare it to a separated-stage training approach. The current ablation studies merely contrast performance with and without the corresponding loss terms, which does not sufficiently justify the effectiveness of joint training. A direct comparison with a sequential training strategy would better support the claims about the benefits of joint training.

**Questions:**

1. What specific method did the authors use to sample nodes from the explanatory graph in the Enhanced Contrastive Learning stage? The text mentions “SimCLR,” but Figure 2 presents “MaskedSimCLR.” Clarifying whether this refers to a direct application of SimCLR to graphs or a modified approach would help clarify the methodology.

2. What is the relationship between $m_e$ in Eq. (9) and $m_{eij}$ in Eq. (6)? If they refer to the same notation, Eq. (9) suggests that optimizing the edge mask directly is possible. How do the authors obtain the ground truth explanation masks in this context?

3. Why do $L_S$ and $L_F$ share the same weight in Eq. (13), given that they represent different optimization objectives? Specifically, $L_S$ focuses on edge explanations, while $L_F$ addresses overall explanations, encompassing both node and edge aspects. Clarifying the rationale for this weighting choice would improve understanding of the training objective.

**Reviewer Confidence:**

4: The reviewer is certain that the evaluation is correct and very familiar with the relevant literature

**Scope:**

4: The work is relevant to the Web and to the track, and is of broad interest to the community

---

### Official Review · Reviewer_ok3g · 2024-12-03

**Novelty:** 4
**Technical Quality:** 4

**Review:**

This paper proposes a self-explainable heterogeneous graph neural network (HGNN) architecture to address the limitation of post-hoc explanation methods, which cannot directly influence predictions. The methodology employs a two-stage design: the first stage generates explanations, and the second stage leverages these explanations to enhance predictions. Additionally, the authors introduce new synthetic datasets for heterogeneous graph explanations.

The paper demonstrates several notable strengths:

- The concept of a self-explaining HGNN is compelling. Post-hoc explanation methods require manual intervention to apply explanations, making the integration of explanation and prediction in this framework an interesting and valuable contribution.
- Although the dataset, *HetBA*, appears to be unpublished at present, its release would represent a significant resource for future research in the domain of heterogeneous graph analysis.

Despite its merits, the paper has certain limitations that require further clarification and discussion:

- One advantage of post-hoc explanation models is their flexibility and adaptability to various tasks and datasets. However, the proposed architecture appears limited in its adaptability. Identifying and discussing scenarios where the method might perform poorly, as well as suggesting possible solutions to address these limitations, would enhance the paper’s utility for readers seeking to apply SEHG in diverse environments.
- While the framework innovatively combines masking explanations with contrastive learning, the paper does not provide assurance or proof of the model’s convergence. This raises concerns about its reliability when applied to datasets beyond those used in the evaluation. Providing a theoretical justification or proof for the model’s convergence would strengthen its technical rigor.

**Questions:**

- Are there any limitations of the proposed method when applied to diverse datasets or tasks?
- If so, is there a conceptual framework or strategy to address these adaptation challenges?
- Can the convergence of the proposed architecture be formally proven or explained?
- Is there a plan to publish the *HetBA* dataset?

**Reviewer Confidence:**

3: The reviewer is confident but not certain that the evaluation is correct

**Scope:**

4: The work is relevant to the Web and to the track, and is of broad interest to the community